# OBJECTFOLDER: A Dataset of Objects with Implicit Visual, Auditory, and Tactile Representations

**Ruohan Gao**      **Yen-Yu Chang**[*]      **Shivani Mall**[*]      **Li Fei-Fei**      **Jiajun Wu**

Stanford University

**Abstract:** Multisensory object-centric perception, reasoning, and interaction have been a key research topic in recent years. However, the progress in these directions is limited by the small set of objects available—synthetic objects are not realistic enough and are mostly centered around geometry, while real object datasets such as YCB are often practically challenging and unstable to acquire due to international shipping, inventory, and financial cost. We present OBJECTFOLDER, a dataset of 100 virtualized objects that addresses both challenges with two key innovations. First, OBJECTFOLDER encodes the visual, auditory, and tactile sensory data for all objects, enabling a number of multisensory object recognition tasks, beyond existing datasets that focus purely on object geometry. Second, OBJECTFOLDER employs a uniform, object-centric, and implicit representation for each object's visual textures, acoustic simulations, and tactile readings, making the dataset flexible to use and easy to share. We demonstrate the usefulness of our dataset as a testbed for multisensory perception and control by evaluating it on a variety of benchmark tasks, including instance recognition, cross-sensory retrieval, 3D reconstruction, and robotic grasping.

**Keywords:** object dataset, multisensory learning, implicit representations

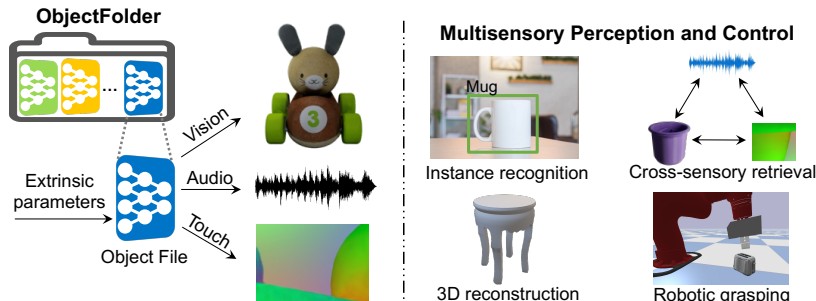

Figure 1: OBJECTFOLDER contains 100 Object Files in the form of implicit neural representations. Querying each Object File with the corresponding extrinsic parameters we can obtain realistic object-centric visual, auditory, and tactile sensory data. The OBJECTFOLDER dataset is a potential testbed for many machine learning and robotics tasks that require multisensory perception.

## 1   Introduction

We perceive the world not as a single giant entity but often through a series of inanimate objects, which exist as bounded wholes and move on connected paths. We interact with these objects through an array of different sensory systems–vision, audition, touch, smell, taste, and proprioception. These multisensory inputs shape our daily experiences: we observe the surroundings to avoid obstacles, hear the doorbell ring to realize the guests have arrived, and touch the fabric to sense its degree of comfort. Cognitive science studies [1, 2] show that both object representation and multisensory perception play a crucial role in human cognitive development.

Object-centric learning has shown great potential lately in visual reasoning [3], representing physical scenes [4], modeling multi-agent interactions [5], and improving generalization to unseen object

---

[*]indicates equal contribution.

5th Conference on Robot Learning (CoRL 2021), London, UK.

compositions [6]. These prior methods, however, often do not model the full spectrum of physical object properties and sensory modes, including their 3D shape, texture, material, sound, and feel. While there has been significant progress by "looking"—recognizing objects based on glimpses of their visual appearance or 3D shape—objects in the world are often modeled as silent and untouchable entities.

We identify that this is due to the lack of rich object datasets. Existing object datasets have contributed to most significant progress in image recognition [7, 8] and shape modeling [9, 10], but other sensory information is minimally considered. Datasets for robotic manipulation research [11, 12] consist of a selected number of real-world household objects, but these objects are expensive and often unstable to acquire due to international shipping and inventory. Moreover, it is nontrivial to virtualize these objects and their multiple sensory modes, limiting their usability for developing embodied AI agents [13, 14, 15, 16, 17, 18, 19, 20, 21].

Our goal is therefore to create a dataset of 3D objects that are 1) easily accessible to the community as a standard benchmark, 2) high-quality in terms of visual textures, and 3) augmented with realistic auditory and tactile sensory data. Towards this end, we introduce OBJECTFOLDER—a multisensory dataset of implicitly represented *Object Files*. The concept of an object file can be traced back to Kahneman *et al.* [22], where it is defined as the sensory information that has been received about the object at a particular location. In a similar spirit, we model and represent each object (or its intrinsics) using an implicit neural representation, which through querying with extrinsic parameters we can obtain images of the object from different viewpoints, impact sounds of the object at each position, and tactile sensing of the object at every surface location. These different sensory data of vision, audio, and touch can be regarded as contemporary multisensory object files.

Specifically, we collect 100 high quality 3D objects from online repositories. For each object, we model its visual appearance at random object pose, lighting condition, and camera viewpoint; we perform modal analysis [23] based on its shape, size, and material type to calculate its characteristic vibration modes for audio simulation; and we simulate its touch readings at all surface locations using DIGIT [24, 25]—a vision-based tactile sensor. Based on recent success on neural implicit representations [26, 27], we design a deep neural network that consists of three sub-networks–VisionNet, AudioNet, and TouchNet, which encode the visual, auditory, and tactile sensory data for each object, respectively. See Fig. 3. Together, they constitute an *Object File* that contains the complete multisensory profile for each object instance. Furthermore, we demonstrate the usefulness of our dataset on four benchmark tasks leveraging multisensory data, including instance recognition, cross-modal retrieval, 3D reconstruction, and robotic grasping.

Our main contributions are threefold: First, we introduce OBJECTFOLDER, a dataset that makes multisensory learning with vision, audio, and touch easily accessible to the research community; Second, all objects in our dataset are compatible to different robotic virtual environments and will be made publicly available as a standard testbed for robotic multisensory learning; Third, we evaluate on a suite of benchmark tasks that require multisensory data to facilitate future work in this direction.

## 2   Related Work

**Object-Centric Datasets**   Many image datasets exist for object recognition, such as ImageNet [7], MS COCO [28], ObjectNet [8], and OpenImages [29]. These datasets consist of 2D images, while OBJECTFOLDER is a dataset of 3D objects. ModelNet [9] and ShapeNet [10] are two large-scale datasets of 3D models, but the collected 3D CAD models either contain no or low-quality visual textures, making them unsuitable for real-world applications. For research on object manipulation, datasets are constructed using real-world household objects such as YCB [11] and BigBIRD [12]. Different from them, our goal is to construct a dataset of 3D objects, where realistic visual, auditory, and tactile sensory data can be easily obtained for each object in virtual environments.

**Implicit Neural Representations**   Coordinate-based multi-layer perceptrons (MLP) have recently been adopted as continuous, memory-efficient implicit representations for a variety of visual signals such as 3D shape [30, 31], scenes [32, 33], and object appearance [26, 27] with the help of classic volume rendering techniques [34]. We also use MLP as a compact neural representation to represent each 3D object. Differently, apart from visual appearance, our mutisensory implicit neural representation also encodes object-centric auditory and tactile data of the object.

**Multisensory Perception for Robotics**   Recent work shows growing interests in using audio or touch in conjunction with vision for robotic tasks. Audio is used to recognize object instances [35]

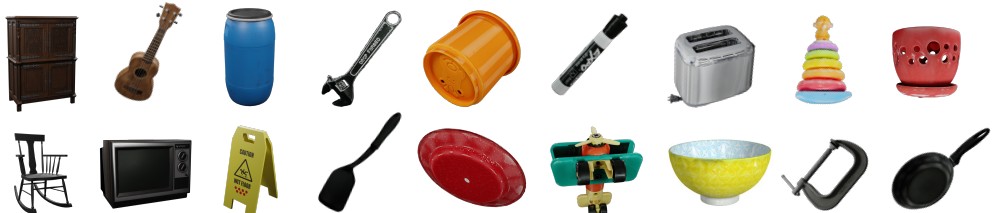

Figure 2: Example objects in OBJECTFOLDER. See Supp. for the visualization of the entire dataset.

or terrain types [36]; estimate the flow and amount of granular materials [37] or liquid height [38] in robotic scooping and pouring tasks; study its synergies with the motion of a robot [39]; and bridge the simulation-to-reality gap for tasks that involve stochastic dynamics [40]. Tactile sensing is more local compared to vision and audio, and it is shown to improve sample efficiency on a peg insertion task [41]; benefit robotic grasping [42, 43, 44]; augment vision to enhance 3D shape reconstruction [45]. Different from the above work that focuses on a particular task that leverages either auditory or tactile data, we construct an object-centric dataset of all three modalities, democratizing multisensory learning research using vision, audio, and touch.

## 3 OBJECTFOLDER

We introduce OBJECTFOLDER, a dataset of 100 implicitly represented 3D objects with vision, audio, and touch. As shown in Fig. 3, each object is represented by an *Object File*, a compact neural network that contains all the visual, acoustic, and tactile profiles for the object. Querying it with extrinsic parameters we can obtain visual appearance of the object from different viewpoints, impact sound of the object at each position, and tactile reading of the object at every surface location. In the following, we first describe the source of our objects and the annotations (Sec. 3.1). Then we present the simulation pipeline and how we encode the sensory data using implicit neural representations for vision (Sec. 3.2), audio (Sec. 3.3), and touch (Sec. 3.4), respectively.

### 3.1 Objects

We collect 100 high quality 3D objects from online repositories including: 20 objects from 3D Model Haven[2], 28 objects from the YCB dataset[3], and 52 objects from Google Scanned Objects[4]. We select objects that are of realistic visual textures, approximately homogeneous material property, and with a material type that is mappable to one of the following categories: ceramic, glass, wood, plastic, iron, polycarbonate, and steel. We annotate each object with the material type, which will be used in audio simulation in Sec. 3.3. The dataset contains common household objects of diverse categories such as bowl, mug, cabinet, television, shelf, fork, and spoon. See Supp. for details.

### 3.2 Vision

**Simulation:** We use Blender's Cycles path tracer [46] to render images. For each object, we first normalize it into a unit cube and use a point light source at a random location on a unit sphere. We then render images of the object on a white background from camera viewpoints randomly sampled on a full sphere. See Fig. 2 for some example views of objects in our dataset.

**VisionNet:** After rendering images from different camera viewpoints under varied lighting conditions, we use VisionNet to encode the visual appearance for each object. Following prior work on object-centric neural radiance fields [27], we represent each object as a 7D object-centric neural scattering function whose input is a 3D location $\mathbf{x} = (x, y, z)$ in the object coordinate frame and the lighting condition at that location $(\omega_i, \omega_o)$, where $\omega_i = (\phi_i, \theta_i)$, $\omega_o = (\phi_o, \theta_o)$ denote the incoming and outgoing light directions, respectively. The output is the volume density $\sigma$ and fraction of the incoming light that is scattered in the outgoing direction $\rho = (\rho_r, \rho_g, \rho_b)$. VisionNet approximates this continuous 7D object-centric representation with an MLP network $F_v : (x, y, z, \theta_i, \phi_i, \theta_o, \phi_o) \longrightarrow (\sigma, \rho_r, \rho_g, \rho_b)$ that maps each input 7D coordinate to its correspond-

---

[2]https://3dmodelhaven.com/
[3]http://ycb-benchmarks.s3-website-us-east-1.amazonaws.com/
[4]https://app.ignitionrobotics.org/GoogleResearch/fuel/collections/Google%20Scanned%20Objects

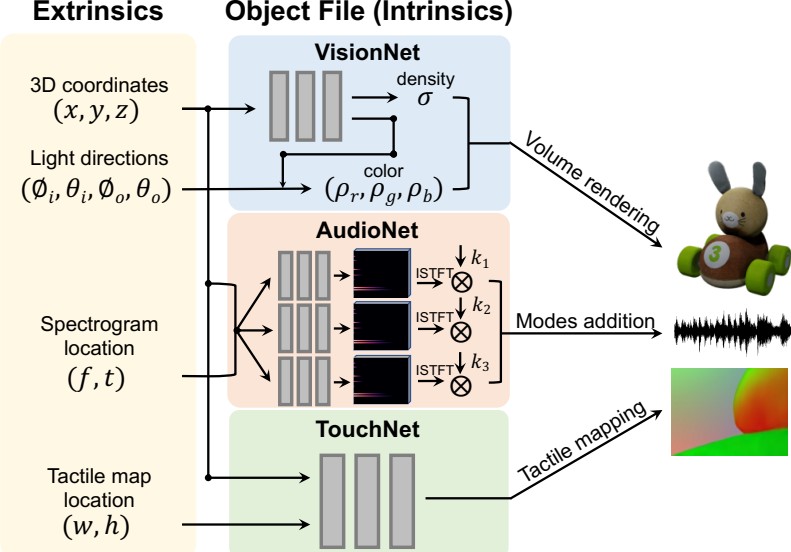

Figure 3: Each *Object File* implicit neural representation contains three sub-networks–VisionNet, AudioNet, and TouchNet, which through querying with the corresponding extrinsic parameters we can obtain the visual appearance of the object from different views, impact sounds of the object at each position, and tactile sensing of the object at every surface location, respectively.

ing volume density and fraction coefficient for each RGB channel. The amount of light scattered at a point $\mathbf{x}$ can be obtained as follows:

$$L_s(\mathbf{x}, \omega_\mathbf{o}) = \int_S L(\mathbf{x}, \omega_i) f_\rho(\mathbf{x}, \omega_\mathbf{i}, \omega_\mathbf{o}) d\omega_\mathbf{i}, \tag{1}$$

where $S$ is a unit sphere, $L(\mathbf{x}, \omega_i)$ denotes the amount of light scattered at point $\mathbf{x}$ along direction $\omega_i$, and $f_\rho$ evaluates the fraction of light incoming from direction $\omega_\mathbf{i}$ at the point that scatters out in direction $\omega_\mathbf{o}$. Then we use classic volume rendering [34] to render the color of any ray passing through the object. The expected color $C(\mathbf{r})$ of camera ray $\mathbf{r}(t) = \mathbf{x}_0 + t\omega_\mathbf{o}$ can be obtained as follows:

$$C(\mathbf{r}) = \int_{t_n}^{t_f} T(t)\sigma(\mathbf{r}(t))L_s(\mathbf{r}(t), \omega_\mathbf{o})dt, \text{ where } T(t) = \exp(-\int_{t_n}^{t} \sigma(\mathbf{r}(s))ds. \tag{2}$$

$T(t)$ denotes the accumulated transmittance along the ray from $t_n$ to $t$; $t_n$ and $t_f$ are the near and far integration bounds; and $\sigma(\mathbf{r}(t))$ denotes the volume density at location $\mathbf{r}(t)$. Similar to [26], we also use stratified sampling, positional encoding, and hierarchical volume sampling to increase rendering quality and efficiency. See [26, 27] for details.

### 3.3 Audio

**Simulation:** The goal of audio simulation is to realistically model the impact sound at each position of the object based on its shape, size, material type, external force, and the contact location. Following the standard approach in engineering and acoustics, we use linear modal analysis for physics-based rigid-body sound synthesis [23, 47].

Firstly, we convert the surface mesh of an object into a volumetric hexahedron mesh, which represents the original shape with $N$ voxels. The key to modal analysis is to solve the following linear deformation equation for a 3D linear elastic dynamics model:

$$\mathbf{M}\ddot{\mathbf{u}} + \mathbf{C}\dot{\mathbf{u}} + \mathbf{K}\mathbf{u} = \mathbf{f}, \tag{3}$$

where $\mathbf{u} \in \mathbb{R}^{3N}$ denotes the displacement of elemental nodes in 3D; $\mathbf{M}, \mathbf{K} \in \mathbb{R}^{3N \times 3N}$ denote the mass and stiffness matrices of the system; $\mathbf{C} = \alpha\mathbf{M} + \beta\mathbf{K}$ stands for Rayleigh damping; and $\mathbf{f} \in \mathbb{R}^{3N}$ represents the external force applied to the object that stimulates the vibration. Based on the material type of the object, we obtain its density value $\rho$, Young's Modulus $E$, Poisson's ratio $r$, and Rayleigh damping parameters $\alpha$ and $\beta$. The scale of the object and material parameters $\rho, E, r$

are used to build the mass and stiffness matrices $\mathbf{M}$ and $\mathbf{K}$. See Supp. for how we map each material type to these parameters.

The above equation can be decoupled into the following form through generalized eigenvalue decomposition $\mathbf{KU} = \mathbf{\Lambda MU}$:

$$\ddot{\mathbf{q}} + (\alpha\mathbf{I} + \beta\mathbf{\Lambda})\dot{\mathbf{q}} + \mathbf{\Lambda q} = \mathbf{U^T f}, \tag{4}$$

where $\mathbf{u} = \mathbf{Uq}$ and $\mathbf{\Lambda}$ is a diagonal matrix. The solution to the above equation is a bank of damped sinusoidal waves with each wave representing a mode signal:

$$q_i = a_i e^{-c_i t} \sin(2\pi\omega_i t + \theta_i), \tag{5}$$

where $\omega_i$ is the frequency of the mode, $c_i$ is the damping coefficient, $a_i$ is the excited amplitude, and $\theta_i$ is the initial phase. Assuming the object is stationary originally, we get the following solution:

$$\theta_i = 0, \quad c_i = \frac{1}{2}(\alpha + \beta\lambda_i), \quad \omega_i = \frac{1}{2\pi}\sqrt{\lambda_i - c_i^2}. \tag{6}$$

$\lambda_i$ represents the $i_{th}$ mode eigenvalue. We simulate the vibration modes excited by unit forces $\mathbf{f_1}, \mathbf{f_2}, \mathbf{f_3}$ for all vertexes and use the obtained modes signals for training AudioNet as detailed next.

**AudioNet:** As shown in Fig. 3, for each force direction, we train a separate branch to encode the corresponding modes signals at all vertexes. We obtain the audio spectrogram[5] for each modes signal and train a MLP to predict the complex spectrogram of dimension $F \times T \times 2$, where $F$ and $T$ are the frequency and time dimensions. Particularly, each branch takes the spatial coordinates $(x, y, z)$[6] and the spectrogram location $(f, t)$ as input, and predict the real and imaginary part of the complex number for every location in the audio spectrogram. The modes signal can then be recovered from the spectrogram using inverse short-time Fourier transform (ISTFT). We also use positional encoding as in VisionNet.

During test time, we directly use vibration modes for sound synthesis following [47]. Any external force at a vertex can be decomposed into a linear combination of unit forces along the three orthogonal directions: $\mathbf{f} = k_1\mathbf{f_1} + k_2\mathbf{f_2} + k_3\mathbf{f_3}$. The amplitudes of modes $\mathbf{a}$ excited by $\mathbf{f}$ can be decomposed into a linear combination of the amplitudes excited by the unit force: $\mathbf{a} = k_1\mathbf{a_1} + k_2\mathbf{a_2} + k_3\mathbf{a_3}$, where $\mathbf{a_1}, \mathbf{a_2}, \mathbf{a_3}$ denote the amplitudes of modes excited by unit forces $\mathbf{f_1}, \mathbf{f_2}, \mathbf{f_3}$ for a given vertex. Because modal analysis is performed on a volumetric hexahedron mesh, for a vertex in the original polygon mesh, we find its closest four vertexes in the hexahedron and average their modes signals.

Prior work [23, 48] have demonstrated that such physics-based rigid-body sound synthesis pipeline renders sound that transfers well to real scenes and the objects' shape, material, and other auditory attributes can be successfully inferred from the simulated sound. To further demonstrate the realism of our audio simulation, we also perform a user study on acoustic fidelity that asks participants to distinguish between the real and simulated audios. Results show that our simulated audio is preferred by 42% of the total responses, comparing closely to real audio recordings. See Supp. for details.

### 3.4 Touch

**Simulation:** We leverage a state-of-the-art touch simulator TACTO [25] for touch simulation. TACTO is a vision-based touch simulator and uses ray-tracing to simulate high-quality tactile signals. It can simulate realistic rendering for both contact surfaces and shadows with different contact geometry. See [25] for details and a comparison of the simulated and real tactile readings. Particularly, we use DIGIT [24] touch sensor and its corresponding camera, lights, and gel mesh configurations in simulation. We choose DIGIT for its compactness, high-resolution of its representation, and potential applications for robotic in-hand manipulation. Other touch sensors such as Gelsight [49], BioTac [50], or Ominitact [51] can also be potentially used, and the design of the touch sensor to use is orthogonal to our work. To obtain touch simulations, we place each object at rest and move DIGIT to touch each object vertex in vertex normal direction. We constrain the force to range within a small threshold such that the rendered tactile images do not vary significantly with different forces. We use the RGB tactile image that contains the local contact geometry as our touch representation.

**TouchNet:** For each object, we obtain a RGB tactile image of dimension $W \times H \times 3$ that captures the local geometry information for every vertex on the surface of the polygon mesh. TouchNet takes

---

[5]We find predicting spectrogram works better than predicting modes signal directly. We suspect spectrogram is more structured by disentangling time and frequency dimensions, making the network easier to optimize.

[6]For audio simulation, we use vertex spatial coordinates on the hexahedron mesh for modal analysis.

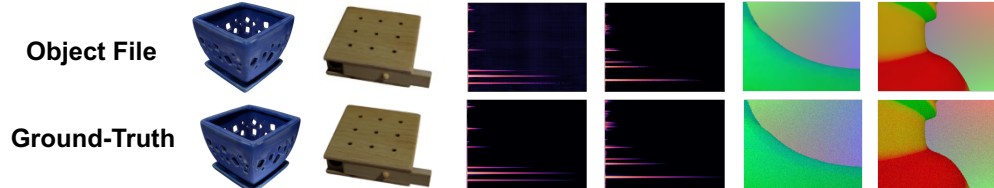

**Object File**

**Ground-Truth**

Figure 4: Comparison of the visual, auditory, and tactile data generated from Object Files and the corresponding ground-truth simulations. Our implicit neural representations accurately encode the sensory data for the objects. See Supp. for more examples and a quantitative comparison.

the spatial coordinate $(x, y, z)$ of the vertex and the spatial location $(w, h)$ in the tactile image as input, and predicts the per-pixel value for the three channels of the tactile RGB image. The other settings are similar to VisionNet and AudioNet.

Fig. 4 shows some examples of the visual, auditory, and tactile sensory data obtained from Object Files. The rendered images, impact sounds, and touch readings well match the ground-truth simulations. See Supp. for a quantitative comparison on the accuracy of the implicit representations. Moreover, using implicit neural representations is much more storage efficient compared to the original data format. Table 1 shows a comparison between the storage size with and without using implicit representations. *Object File* in the form of

|  | Implicit | Original |
|---|---|---|
| Vision | 7.2 MB | $\infty$ |
| Audio | 6.3 MB | 12.6 GB |
| Touch | 2.1 MB | 2.9 GB |

Table 1: Storage space comparison.

an implicit neural network is a much more compact representation of the multisensory data for each object. Note that VisionNet models an object-centric neural scattering function [27] and thus can generalize to any camera viewpoint under arbitrary lighting conditions, while rendering and saving images for all scenarios takes infinite space theoretically.

## 4 Experiments

OBJECTFOLDER is a potential testbed for many perception and control tasks. Next, we evaluate on four benchmark tasks including instance recognition, cross-sensory retrieval, 3D reconstruction, and robotic grasping to demonstrate the usefulness of the dataset and the value of multisensory perception. We show the key results below, and see Supp. for the detailed experiments setup.

### 4.1 Multisensory Instance Recognition

Identifying the object instance that is interacted with is fundamental for many robotic applications. In this task, we want to identify an object based on either its visual appearance, impact sound, or local contact geometry. We are interested in finding out the amount of information that is useful to recognize the object instance from each modality. For the vision modality, we train a ResNet-18 [52] network that takes an RGB image of the object as input and predicts the instance label of the object. The settings are the same for audio and touch except that the input is either a magnitude spectrogram of the impact sound the object generates or a local tactile RGB image from a random location on the object surface.

| Methods | Acc. (%) |
|---|---|
| Chance | 1.00 |
| Vision (V) | 94.8 |
| Audio (A) | 98.3 |
| Touch (T) | 72.4 |
| V + A | 99.5 |
| V + T | 97.2 |
| A + T | 99.0 |
| V + A + T | 99.8 |

Table 2: Results on multisensory instance recognition.

Table 2 shows the results. The accuracy for a random classifier is 1%. The vision classifier achieves high accuracy, because an image of the object normally captures informative appearance cues that distinguish the object instance. Under some camera viewpoints, the vision classifier can mistakenly classify the object. Surprisingly, the audio classifier achieves the best results. This confirms that our audio simulation pipeline well models the shape, size, and material property of the object instance, so that a single impact sound can be used to recognize the object. Touch is good at capturing local geometry of an object, but often does not contain sufficient discriminative cues to recognize the object especially from a single touch. Therefore, the touch classifier performs worse compared to vision and audio. Classifiers with multisensory input are more robust, and combining all three modalities leads to the best result.

## 4.2 Cross-Sensory Retrieval

Cross-sensory retrieval plays a crucial role in machine perception to understand the relationships between different modalities. Fig. 5 shows our framework for learning cross-sensory embeddings for two modalities. Taking cross-sensory learning with vision and audio as an example, we design a two-stream network that takes an RGB image and an audio spectrogram of the same object as input for each stream. The two modalities have disjoint pathways in the early layers to capture the modality-specific features. A final modality-agnostic network is used to map the shared embedding to the label space for training with cross-entropy loss. We map the features of both modalities from the last hidden state to a cross-sensory embedding space through a triplet loss. We either sample an image or audio from another object as the negative example, and pull the embeddings of the same object to be closer in the joint embedding space and push the embeddings from different objects apart. The full model is trained jointly with both the triplet loss and the cross-entropy loss.

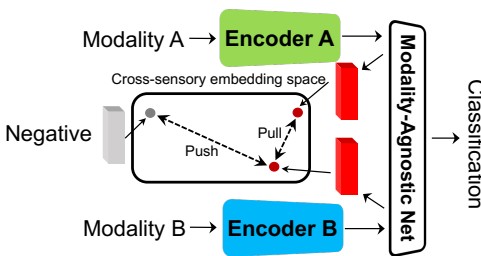

Figure 5: Learning cross-sensory embeddings.

| Input | Retrieved | Chance | CCA [53] | Ours |
|-------|-----------|--------|----------|------|
| Vision | Audio | 0.05 | 0.57 | 0.90 |
| | Touch | 0.05 | 0.24 | 0.50 |
| Audio | Vision | 0.05 | 0.59 | 0.92 |
| | Touch | 0.05 | 0.31 | 0.55 |
| Touch | Vision | 0.05 | 0.29 | 0.48 |
| | Audio | 0.05 | 0.33 | 0.64 |

Table 3: Cross-sensory retrieval results.

Table 3 shows the results for cross-sensory retrieval on the test set, where we retrieve the sensory data of a different modality by querying from the shared embedding space. We use cosine similarity to measure the distances between embeddings and retrieve the sample with the smallest distance. We report mean Average Precision (mAP) which jointly considers the ranking information and the precision, and compare with random retrieval and a state-of-the-art cross-sensory retrieval method based on canonical correlation analysis (CCA) [53]. Our models perform more reliable retrievals, and the results show that the learned cross-sensory embeddings are effective and contain fine-grained information about objects to correctly retrieve sensory data from the matched object. Similar to instance recognition, we observe vision and audio provide more reliable global information about the object compared to touch.

## 4.3 Audio-Visual 3D Reconstruction

Robots often need to build a mental model of the 3D shape of objects based on just a single glimpse or the sounds they make during interaction. In this section, we investigate the potential of 3D reconstruction using either a single image, an impact sound, or their combination. We leverage Occupancy Network [30], which implicitly represents the 3D surface as the continuous decision boundary of a deep neural network classifier, as a testbed for this task. Fig. 6 illustrates our IMAGE+AUDIO2MESH framework for audio-visual 3D shape reconstruction. The image and audio embeddings are fused through a fusion layer into an audio-visual feature, then combined with coordinate-conditioned feature maps through Conditional Batch-Normalization to predict occupancy probabilities $p$. We also test two of its simplified variants IMAGE2MESH, which represents existing single image 3D reconstruction approaches that predict the 3D shape from a single view of the object, and AUDIO2MESH, which performs 3D reconstruction purely from audio.

| Methods | IoU ↑ | Chamfer-$L_1$ ↓ | Normal Consistency ↑ |
|---------|-------|-----------------|----------------------|
| AVERAGE | 0.0675 | 0.1067 | 0.6312 |
| IMAGE2MESH [30] | 0.8809 | 0.0046 | 0.9522 |
| AUDIO2MESH | 0.8729 | 0.0048 | 0.9504 |
| IMAGE+AUDIO2MESH | 0.8906 | 0.0043 | 0.9535 |

Table 4: 3D shape reconstruction results. ↓ lower better, ↑ higher better.

We evaluate on a held out test set with standard metrics: IoU, Chamfer-$L_1$ distance, and Normal Consistency with respect to the ground-truth mesh. Table 4 shows the results. Compared to the

simple AVERAGE baseline that averages the results obtained using the ground-truth mesh of each of the 100 objects as the prediction, we can see that 3D reconstruction from a single image or an impact sound performs much better. Augmenting traditional single image 3d reconstruction with audio achieves the best performance by leveraging the additional acoustic spatial cues. Furthermore, as shown in Fig. 7, we apply our IMAGE2MESH model trained on objects in OBJECTFOLDER to real-world images. Our model generalizes reasonably well to real image, demonstrating the realism of the objects in our dataset. The last column shows a typical failure case, where the teapot image is very visually different from objects in our dataset.

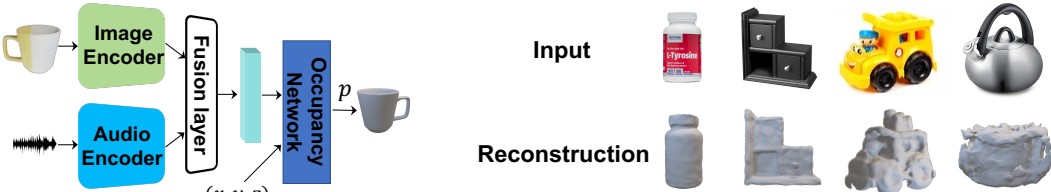

Figure 6: IMAGE+AUDIO2MESH framework for 3D reconstruction.

Figure 7: 3D shape reconstruction using real-world images. Last column shows a typical failure case.

## 4.4 Robotic Grasping with Vision and Touch

Next we show how we combine vision and touch for prediction of grasp stability. The goal is to use a robotic arm to successfully grasp and hold an object between the robot's left and right fingers. For touch data, we infer the RGB tactile image from the corresponding Object File based on the grasp contact positions of the left and right grippers on the object; For vision data, we obtain images from an externally mounted camera's viewpoint at the grasping moment. We label the data as either "Success" or "Failure" based on whether the object can be held between the robot's fingers after being lifted. Fig. 8 shows an example of both a successful grasp and a failure case.

We collect data for 10 objects from OBJECTFOLDER, and then train binary classifiers under different sample sizes to predict the grasp outcome based on either the vision signal, the touch signal, or their combination. This experiment is similar to that of [44, 25] except that we obtain touch readings directly by querying our implicit representation networks. Fig. 9 shows the results. We observe that it takes significantly less amount of data to reach high accuracy when learning from touch compared to vision. Leveraging both visual and tactile data achieves the best accuracy. Additionally, we train a policy for touch-based robotic grasping using TRPO [54]. We achieve a success rate of 75.5% during testing, while that of a random grasping policy is 53.0%. To further demonstrate the potential of using our dataset on robot manipulation tasks, we also perform an experiment on an object manipulation task (reach) using Meta-World [55] with three of our objects (cup, bowl, dice). We achieve 100% success rate for each object. See Supp. for details.

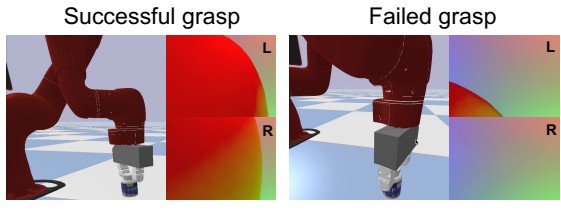

Figure 8: Examples of a successful grasp and a failed grasp. L: left finger, R: right finger.

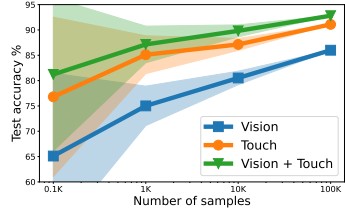

Figure 9: Grasp stability prediction.

## 5 Conclusion

We presented OBJECTFOLDER, a dataset of objects with implicit visual, auditory, and tactile representations. It will facilitate research with vision, audio, and touch, and enable multisensory simulation of objects in robotic virtual environments. Through evaluation on an array of benchmark tasks, we show the usefulness of our dataset for both perception and control. As future work, we plan to explore richer object states with more accurate and fine-grained physics, additional sensory modalities, and textual descriptions.

**Acknowledgements:** We thank Michelle Guo, Xutong Jin, Shaoxiong Wang, Huazhe Xu, and Samuel Clarke for helpful discussions on experiments setup and Michael Lingelbach and Jingwei Ji for suggestions on paper drafts. The work is in part supported by Toyota Research Institute (TRI), Samsung Global Research Outreach (GRO) program, ARMY MURI grant W911NF-15-1-0479, NSF CCRI #2120095, Amazon Research Award (ARA), Autodesk, Qualcomm, and Stanford Institute for Human-Centered AI (HAI).

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
