# OpenReview forum: "ObjectFolder: A Dataset of Objects with Implicit Visual, Auditory, and Tactile Representations"
_robot-learning.org/CoRL/2021/Conference — CoRL2021 Poster_

### Official Review · Reviewer_pXzD · 2021-07-21

**Originality:** Very Good
**Technical Quality:** Excellent
**Clarity Of Presentation:** Excellent
**Impact:** 4

**Recommendation:**

Strong Accept: I recommend accepting the paper and will argue for my recommendation even if other reviewers hold a different opinion.

**Summary:**

This paper presents a dataset of 100 virtual objects with visual, auditory, and tactile sensory input for learning multi-modal object representations. The dataset is constructed from scanned real-world objects available from ModelHaven, YCB, and Google Scanned Objects. The audio is synthesized through rigid-body sound simulation and touch is simulated with TACTO for emulating a DIGIT sensor. To investigate the usefulness of multimodal representations, the paper presents a model for learning neural representations with three subnetworks – VisionNet, AudioNet, and TouchNet. Evaluations on downstream tasks such as instance recognition, cross-sensory retrieval, and grasping, show that using multimodal representations result in significant performance gains over simple unimodal or bimodal baselines.

**Issues:**

Minor comments:
- From the supplementary video, the center-of-mass (COM) of Google Scanned Objects seem to be incorrect, leading to wobbly objects that never fall down. This is a known issue with Google Scanned Objects where the COM is at the bottom of the object rather than it’s geometric center or true center. Accordingly, you probably need to re-run the grasping experiments with the corrected COMs.
- How many training images were used to train the NeRF? How long does it take to train a NeRF per object?


**Reviewer Expertise:**

Excellent: Expert knowledge on the topic of the paper

**Strengths And Weaknesses:**

Strengths:
- This paper makes an important resource contribution to robotics. Most robot learning methods heavily rely on vision for perception. There has been overwhelming evidence that perception for manipulation is inherently multimodal and to learn robust and dexterous skills, robotic systems need to go beyond just visual perception.
- Benchmarking and reproducibility are crucial for evaluating progress in various robotic domains. Having a standardized set of objects for multimodal perception, particularly in a simulation, is very beneficial for benchmarking future works in detection, grasping, and dexterous manipulation. To the best of my knowledge, this is the first benchmark that provides all three modalities – vision, audio, and tactile, in a simulated setup.
- The evaluations on downstream tasks are insightful. In all cases from detection to grasping, the multimodal representations outperform unimodal and bimodal baselines.

Weaknesses:
- One concern, although relatively minor, is the quality of the audio and tactile simulation and their fidelity to real-world sensors. While the visual rendering of scanned objects is near photorealistic, the audio simulation uses rigid-body simulation to emulate audio sensors. For this simulation, each object is modelled with a single material since individual parts of the scanned objects are unknown. This could be an issue for some objects, for example, striking a hammer on the plastic handle might produce a different sound than striking it on the metallic head. This might also explain why the Audio-only baseline in Table 2 performs so well. Similarly, the simulated tactile sensor uses the geometric structure of scanned objects which could have some imperfections and sharp edges from 3D reconstruction. Ideally, both these issues can be addressed by evaluating the simulated sound and tactile inputs against real-world sensor data. This can be done by picking a few objects from the dataset, collecting data with real-world objects, and comparing it against the simulated audio and tactile sensors.
- Another concern, noticeable in the supplementary video, is that the dataset contains a lot of cups or cup-like objects of cylindrical shapes. Was this a specific design choice? Or a symptom of what was available/feasible for robot grasping? Since the audio and tactile inputs are automatically simulated, is there anything preventing the dataset from containing more than 100 objects? Perhaps all of Google Scanned Objects and ModelHaven objects? Benchmarks need to be careful not to induce biases in the dataset that might result in models overfitting or exploiting biased data without learning how to solve the task.


**Summary Of Recommendation:**

This paper makes an important resource contribution with a multimodal dataset of virtual objects. Experimental evaluations on various downstream robotics tasks indicate that multimodal representations are crucial for going the last mile in detection, grasping, and related tasks.

**Post Rebuttal**
I am happy with the authors' response and followup experiments, so I am keeping my original score. I'd encourage the authors to add more diverse objects to the dataset, if possible.

---

> ### Author Response · Authors · 2021-08-29
> **Response to Reviewer pXzD**
>
> Thank you for the constructive feedback!
>
> **(a) quality of the audio and tactile simulation and their fidelity to real-world sensors; a single material … could be an issue for some objects; the simulated tactile sensor … could have some imperfections and sharp edges.**
>
> Our audio simulation realistically models the impact sound of each object based on its shape, size, material type, external force, and contact location (L133-36). Prior work (Zhang et al. NeurIPS’17,  Ren et al. TOG’13) has shown that such physics-based rigid-body sound synthesis pipeline renders sound that transfers well to real scenes, and the objects’ shape, material, and other auditory attributes can be successfully inferred from the simulated sound. To further show the realism of our audio simulation, we have performed a new user study that asks listeners to distinguish between the real and simulated audios for some everyday objects (e.g., a ceramic mug, a steel bowl). Results show that our simulated audio is preferred by 42% of the total responses, comparing closely to real audio recordings. See Sec. D in Supp. for details.
>
> We select objects that are of approximately homogeneous material property (L103) because our physics-based rigid-body sound synthesis pipeline assumes a single material type for each object. There are some approximations made for a few objects of non-homogeneous material such as a bell with a wood handle, a fork with a plastic handle, etc. Sound simulation with multiple or complex material types is a hard and unsolved problem on its own (Liu & Manocha arXiv’21, Sterling et al. TVCG’19), which we leave as future work to improve the realism of the sound simulation for these objects.
>
> Our tactile simulation is based on SOTA tactile simulator—TACTO (Wang et al. NeurIPS’20), which uses ray-tracing to simulate high-quality tactile signals (L171-75). It can simulate realistic rendering for both contact surfaces and shadows with different contact geometry. See Fig. 9 in the TACTO paper for some examples of how the simulated tactile images well match the tactile readings of real objects.
>
> **(b) a lot of cups or cup-like objects:**
>
> These cups or cup-like objects come from the shape items in the YCB dataset. These objects are useful for object manipulation tasks that require shape analysis. For example, these objects are of the same material type and are very visually similar. The differences in shape make their acoustic characteristics different in modal analysis, making these objects distinguishable from each other in multisensory object identification and manipulation tasks.
>
> **(c) anything preventing the dataset from containing more than 100 objects?**
>
> None. It would be straightforward and interesting to expand our dataset to cover a wider range of objects (e.g., more Google Scanned Objects). In this first version, we have selected the 100 objects from existing repositories that contain realistic visual textures, approximately homogeneous material properties, and with a material type that is mappable to a known category (L101-07). The raw sensory data for each object takes large storage space (Table 1), and therefore we focus on 100 high-quality objects in our dataset. Using implicit representations to represent each object’s visual textures, acoustic simulations, and tactile readings makes the dataset flexible to use and easy to share.
>
> **(d) the center-of-mass (COM) of Google Scanned Objects seem to be incorrect?**
>
> Thanks for pointing this out. We have fixed this issue and updated the results in the paper. See Supp. for the new video where we have fixed the COM of the wobbly objects.
>
> **(e) How many training images were used to train NeRF?**
>
> 500 images for each object are used for training. See L32-36 in Supp. for details.

---

> > ### Comment · Reviewer_pXzD · 2021-09-02
> > **Thanks for response.**
> >
> > Thank you for the detailed response. As stated above, I am keeping my original score.

---

> > > ### Author Response · Authors · 2021-09-03
> > > **Thank you!**
> > >
> > > Thank you so much for taking the time to read our response! We are very grateful for your valuable feedback and strong support for our work.

---

### Official Review · Reviewer_arko · 2021-07-24

**Originality:** Good
**Technical Quality:** Very Good
**Clarity Of Presentation:** Excellent
**Impact:** 4

**Recommendation:**

Strong Accept: I recommend accepting the paper and will argue for my recommendation even if other reviewers hold a different opinion.

**Summary:**

The authors present ObjectFolder, a dataset containing 100 objects with implicit visual, audio and tactile representations. A novel method is proposed for representing the audio signals implicitly. Once this dataset is built, the implicit functions are evaluated in a series of passive benchmarks measuring 3D reconstruction quality, cross-modal instance recognition/retrieval and grasping stability prediction in simulation.

**Issues:**

See above.

**Reviewer Expertise:**

Excellent: Expert knowledge on the topic of the paper

**Strengths And Weaknesses:**

First of all, I must say, this paper was a joy to read and is very original. The approach is very novel and could potentially be useful for the robotics community which has traditionally focused on explicit data representations with mixed results.

VisionNet and TouchNet seem to have limited contributions since they are mainly inspired from NERF. The representation for AudioNet on the other hand seems to be a novel contribution.

The biggest concern for the paper is that there is no comparison to explicit methods. For instance, a direct comparison to the network architecture in Calandra et al. [CoRL 2017] would be insightful for the grasp stability experiment. Without this comparison, as a reader I was still wondering why a implicit representation is functionally necessary over an explicit one.

Also it is challenging to understand if the implicit representation transfers well to real world tasks since the results were only passively tested (i.e. without contact-rich manipulation) or in simulation.

Table 3 is very insightful. It shows that vision and audio are good modalities to capture global information (like the object instance or its category) compared to tactile (which mainly has local geometric information).

**Summary Of Recommendation:**

I support to accept this paper in its current form. Despite my reservations on the potential impact and widespread adoption of the dataset in the community, there are definitely novel contributions in the idea and architecture.

---

> ### Author Response · Authors · 2021-08-29
> **Response to Reviewer arko**
>
> Thank you for the constructive feedback!
>
> **(a) comparison to explicit methods, a direct comparison to Calandra et al. 2017?**
>
> Thanks for the suggestion. We wish to clarify that Calandra et al. CoRL’17 proposed a network architecture, which can leverage both the explicit and implicit representations as input; instead, our ObjectFolder is a new set of implicitly parameterized objects, which can be used together with Calandra et al. or any other architectures.
>
> To demonstrate the effectiveness of our implicit representations, we have performed a new experiment that directly uses the explicit tactile representations (ground-truth simulations) as input for the grasp stability prediction experiment. All other settings are the same except the input to the network to directly compare the implicit and explicit methods in an apples-to-apples manner. Figure 5 in Supp. shows the results. Object Files in the form of compact implicit representations achieve similar results as using the explicit tactile representations while having the advantage of being flexible to use and easy to share.
>
> **(b) why an implicit representation is functionally necessary over an explicit one:**
>
> Table 1 and L194-206 show that using implicit representations is much more storage efficient than the original data format for each sensory modality, making the dataset easy to share. Object File in the form of an implicit neural network compactly encodes the multisensory data for each object, which through querying with the corresponding extrinsic parameters we can easily obtain the visual appearance of the object from different views, impact sounds of the object at each position, and tactile sensing of the object at every surface location, respectively.
>
> **(c) if the implicit representation transfers well to real-world tasks**
>
> As pointed out by R3, ObjectFolder is the first benchmark that provides all three modalities—vision, audio, and touch in a simulated setup, which is very beneficial for benchmarking future works in detection, grasping, and dexterous manipulation. We have evaluated four benchmark tasks to show the usefulness of our dataset for multisensory perception and control, including instance recognition, cross-sensory retrieval, 3D reconstruction, and robotic grasping. While our dataset is based on simulation, the results show that the learned models already benefit from it when transferring to real-world data for single image 3D reconstruction (Fig. 7), indicating the realism of our objects.
>
> Nonetheless, it will be interesting future work to directly perform Sim2Real object transfer—learning with virtualized objects in simulation and then transferring to real-world counterparts for robotics tasks. Due to COVID-19, unfortunately we were not able to get access to these real objects or robots easily; however, per request, to further demonstrate the potential of using our dataset for real robotic manipulation tasks, we perform an additional experiment on an object manipulation task using Meta-World (Yu et al. CoRL'19): reach. We follow the standard MT1 setting in the Meta-World paper and train a policy for reach with three of our objects (cup, bowl, dice). We have achieved a 100% success rate for each object, demonstrating the usefulness of our objects for robotic manipulation. See page 8 in the revised paper and Sec. J in Supp. for details.

---

> > ### Comment · Reviewer_arko · 2021-09-02
> > **Thanks for the comments**
> >
> > Thanks for the response. I am still concerned regarding the applicability of this implicit representation on a real robot task such as grasping (without a proper experiment). Nonetheless, I keep my original score.

---

> > > ### Author Response · Authors · 2021-09-03
> > > **Thank you!**
> > >
> > > Thank you so much for taking the time to read our response! We have evaluated four benchmark tasks and added one additional experiment on object manipulation in simulation, which demonstrates the potential of using our dataset (implicit representations) for real robotic manipulation tasks. Our Sim2Real 3D reconstruction experiments also indicate the realism of our objects. Due to time and COVID constraints, we were not able to directly perform robotic grasping on a real robot in rebuttal, which we leave as future work.
> > >
> > > Again, we are very appreciative of your valuable input and strong support for our work.

---

### Official Review · Reviewer_BbKQ · 2021-07-26

**Originality:** Fair
**Technical Quality:** Good
**Clarity Of Presentation:** Good
**Impact:** 3

**Recommendation:**

Weak Accept: I recommend accepting the paper, but will not argue for my recommendation if the majority of other reviewers have a different opinion.

**Summary:**

The paper aimed to tackle the problem that Multisensory object-centric perception is limited by the small set of objects available. It presented OBJECTFOLDER, a dataset of 100 virtualized objects with implicit visual, auditory, and tactile representations. The usefulness of the proposed dataset was evaluated on an array of benchmark for both perception and control.


**Issues:**

- I have doubts that whether such dataset can well represent auditory, and tactile attributes of the real objects. Therefore, additional experimental analysis need to be presented to show that the multisensory data can well represent object attributes.

- The paper mentioned that the rendered images, impact sounds, and touch readings well match the ground-truth simulations. It needs more experimental analysis to support this conclusion.

- According to the focus of the paper, it needs to redesign the experiments to prove the validity of the dataset. In particular, some experiments need to be added to verify effectiveness of the dataset for robot perception and learning task in real scenarios.

- The dataset is evaluated on four different benchmark tasks. This resulted in that experimental details are ignored and the experimental analysis is not sufficient. I think that focusing on one benchmark task can improve the clarity and completeness of the paper.


**Reviewer Expertise:**

Good: General knowledge of the area

**Strengths And Weaknesses:**

Strengths:
- For robotic applications, it is very expensive to obtain large-scale multisensory dataset, especially auditory and tactile data. The limitations have long been noticed. This paper proposed a novel method to obtain visual, auditory, and tactile sensory data of objects from virtual environments, which is well motivated.

- The paper introduced OBJECTFOLDER, a dataset of 00 virtualized objects that makes multisensory learning with vision, audio, and touch easily accessible to the research community.

- OBJECTFOLDER employs a uniform, object-centric, and implicit representation for each object’s visual textures, acoustic simulations, and tactile readings, making the dataset flexible to use and easy to share.

Weaknesses:
- Although the dataset contains100 object-centric visual, auditory, and tactile sensory data, the reviewer has doubts that whether such dataset can well represent auditory, and tactile attributes of the real objects. This should be discussed appropriately.

- The paper mentioned that the rendered images, impact sounds, and touch readings well match the ground-truth simulations. If only based on the results in Figure 4, this conclusion may be too arbitrary. It needs more experimental analysis to support this conclusion.

- The paper claims the contribution that all objects in the proposed dataset are platform agnostic, which may not be a valid contribution. As a testbed, it should be the basic requirement.

- The paper adopted a variety of benchmark tasks to demonstrate the validity of the data set, which is unconvincing. The paper should verify whether the dataset is valid for robot perception and learning task in real-life scenarios.


**Summary Of Recommendation:**

Although the method of obtaining visual, auditory, and tactile sensory data of objects from virtual environments is helpful for robotic multimodal learning, the presented experimental analysis did not effectively demonstrate the contribution of the paper. The paper needs to be further modified by adding additional experiments to improve the quality. Therefore, it is not ready for acceptance.

---

> ### Author Response · Authors · 2021-08-29
> **Response to Reviewer BbKQ**
>
> Thank you for the constructive feedback!
>
> **(a) whether such dataset can well represent auditory and tactile attributes of the real objects:**
>
> Our audio simulation realistically models the impact sound of each object based on its shape, size, material type, external force, and contact location (L133-36). Prior work (Zhang et al. NeurIPS’17,  Ren et al. TOG’13) has shown that such physics-based rigid-body sound synthesis pipeline renders sound that transfers well to real scenes, and the objects’ shape, material, and other auditory attributes can be successfully inferred from the simulated sound. Based on your feedback, in order to further demonstrate the realism of our audio simulation, we have performed a new user study that asks listeners to distinguish between the real and simulated audios for some everyday objects (e.g., a ceramic mug, a steel bowl). Results show that our simulated audio is preferred by 42% of the total responses, comparing closely to real audio recordings. See Sec. D in Supp. for details.
>
> Our tactile simulation is based on the SOTA tactile simulator—TACTO (Wang et al. NeurIPS’20), which uses ray-tracing to simulate high-quality tactile signals (L177-83). It can simulate realistic rendering for both contact surfaces and shadows with different contact geometry. See Fig. 9 in the TACTO paper for some examples of how the simulated tactile images well match the tactile readings of real objects.
>
> **(b) the rendered images, impact sounds, and touch readings well match the ground-truth simulations (GT)—more experimental analysis to support this conclusion:**
>
> The comparisons in Fig. 4 in main and Fig. 3 in Supp. illustrate the high quality of our implicit representations qualitatively. Per request, we have performed a new experimental analysis to compare the sensory data obtained from Object Files with GT quantitatively: 1) For vision, we render 500 images for each object and compute the average PSNR w.r.t. GT; 2) For audio, we obtain 500 impact sounds for each object and compute the average PSNR of the spectrograms w.r.t. GT; 3) For touch, we render 500 tactile images at different surface locations and compute the average PSNR w.r.t. GT. For each modality, we compare with a baseline where GT is corrupted by negligible white Gaussian noise (AWGN, with mean=0 and variance=0.05) as a reference. The table below shows the results. We can see that the sensory data encoded by our implicit representation networks achieve very high PSNR values (in dB), and compare favorably against the baseline.
>
> ||Implicit w.r.t. GT|AWGN w.r.t. GT|
> |:-:|:-:|:-:|
> |Vision|39.8|32.0|
> |Audio|53.6|26.2|
> |Touch|29.4|24.2|
>
> **(c) validity of the dataset, ...experiments need to be added to verify the effectiveness of the dataset for robot perception and learning task in real scenarios:**
>
> ObjectFolder is the first benchmark that provides all three modalities—vision, audio, and touch in a simulated setup, which is very beneficial for benchmarking future works in detection, grasping, and dexterous manipulation (R-pXzd). We have evaluated four benchmark tasks to show the usefulness of our dataset for multisensory perception and control, including instance recognition, cross-sensory retrieval, 3D reconstruction, and robotic grasping. While our dataset is based on simulation, the results show that the learned models already benefit from it when transferring to real-world data for single image 3D reconstruction (Fig. 7), indicating the realism of our objects.
>
> Nonetheless, it will be interesting future work to directly perform Sim2Real object transfer—learning with virtualized objects in simulation and then transferring to real-world counterparts for robotics tasks. Due to COVID-19, unfortunately we were not able to get access to these real objects or robots easily; however, per request, to further demonstrate the potential of using our dataset for real robotic manipulation tasks, we have performed an additional experiment on an object manipulation task using Meta-World (Yu et al. CoRL'19): reach. We followed their standard MT1 setting, and trained a policy for reach with three of our objects (cup, bowl, dice). We achieved a 100% success rate for each object, demonstrating the usefulness of our objects for robotic manipulation. See Sec. J in Supp. for details.
>
> **(d) experimental details and analysis …, … focusing on one task can improve the clarity and completeness of the paper:**
>
> The experimental setup details for all four tasks are in Sec. I of Supp. ObjectFolder is a potential testbed for many perception and robotics tasks. Our goal is to demonstrate this dataset is generally useful for multisensory perception and control instead of achieving SOTA performance on one particular task. Therefore, the dataset is mainly evaluated on four different benchmark tasks (L208-11). Experiments across these tasks show that using multimodal representations results in significant performance gains over simple unimodal or bimodal baselines.

---

> > ### Author Response · Authors · 2021-09-03
> > **Thank you!**
> >
> > Dear Reviewer BbKQ,
> >
> > Many thanks again for your valuable feedback on our work! If you have any additional questions after reading our response, please don't hesitate to let us know. We are very grateful for your input to strengthen our work.
> >
> > Thank you!
> > ObjectFolder Team

---

> > > ### Comment · Reviewer_BbKQ · 2021-09-03
> > > **Thanks for the response.**
> > >
> > > Thanks for the response. The response has answered some of my concerns and the score has been updated. It is true that the idea of having a multimodal object dataset is good. And I'm still curious about how the dataset can benefit real robotic applications.

---

> > > > ### Author Response · Authors · 2021-09-03
> > > > **Thanks!**
> > > >
> > > > Thank you so much for taking the time to read our response! We are glad to know that your concerns are addressed. We have evaluated four benchmark tasks and added one additional experiment on object manipulation in simulation, which demonstrates the potential of using our multisensory object dataset for real robotic applications. Our Sim2Real 3D reconstruction experiments also indicate the realism of our objects. Due to time and COVID constraints, we were not able to directly perform real robot experiments in rebuttal, which we leave as future work.
> > > >
> > > > Again, we are very appreciative of your valuable input and support for our work.

---

### Meta-Review · Area_Chair_hdCq · 2021-08-11

**Recommendation:** Accept (Poster)
**Confidence:** 4

**Metareview:**

This paper proposes a new dataset of 100 synthetic objects with visual, auditory, and tactile sensory features for learning multi-modal object representations. The dataset combines objects from the YCB dataset, the ModelHaven, and the Google Scanned Objects. The benefits of the proposed dataset as a testbed for multisensory perception and control are demonstrated by evaluating it on a variety of benchmark tasks, including instance recognition, cross-sensory retrieval, 3D reconstruction, and robotic grasping.
The reviewers agree that the proposed dataset can be useful for pushing ahead the state of the art in robotic grasping and manipulation. A key issue however is the realism of the dataset, which is not clear from this paper. The authors should demonstrate the benefits of using this dataset on real robotic manipulation tasks.

---

> ### Author Response · Authors · 2021-08-28
> **Summary**
>
> Thank you, AC and all reviewers, for all your valuable input. We are glad to see that R-arko finds our approach very novel and useful for the robotics community; R-pXzD thinks the paper makes an important resource contribution to robotics; R-BbKQ finds our method novel and helpful for robotic multimodal learning, but has some concerns about the experimental analysis. Reviewers’ questions are mainly about the realism of our auditory and tactile simulations and the benefits of using this dataset on real robotic manipulation tasks. We address them in our separate responses accordingly.

---

### Decision · Program_Chairs · 2021-09-13

**Decision:**

Accept (Poster)

**Comment:**

This paper proposes a new dataset of 100 synthetic objects with visual, auditory, and tactile sensory features for learning multi-modal object representations. The dataset combines objects from the YCB dataset, the ModelHaven, and the Google Scanned Objects. The benefits of the proposed dataset as a testbed for multisensory perception and control are demonstrated by evaluating it on a variety of benchmark tasks, including instance recognition, cross-sensory retrieval, 3D reconstruction, and robotic grasping.
The reviewers agree that the proposed dataset can be useful for pushing ahead the state of the art in robotic grasping and manipulation. A key issue however is the realism of the dataset, which is not clear from this paper. The authors should demonstrate the benefits of using this dataset on real robotic manipulation tasks.